# Tuning of the Titanium Oxide Surface to Control Magnetic Properties of Thin Iron Films

**DOI:** 10.3390/ma16010289

**Published:** 2022-12-28

**Authors:** Juliusz Chojenka, Arkadiusz Zarzycki, Marcin Perzanowski, Michał Krupiński, Tamás Fodor, Kálmán Vad, Marta Marszałek

**Affiliations:** 1Institute of Nuclear Physics Polish Academy of Sciences, PL-31342 Krakow, Poland; 2Institute for Nuclear Research, Hungarian Academy of Science, Bem tér 18/C, H-4026 Debrecen, Hungary

**Keywords:** iron thin films, interphase magnetic exchange interaction, interface oxidation, nanoporous titanium oxide

## Abstract

We describe the magnetic properties of thin iron films deposited on the nanoporous titanium oxide templates and analyze their dependance on nanopore radius. We then compare the results to a continuous iron film of the same thickness. Additionally, we investigate the evolution of the magnetic properties of these films after annealing. We demonstrate that the *M*(*H*) loops consist of two magnetic phases originating from the iron layer and iron oxides formed at the titanium oxide/iron interface. We perform deconvolution of hysteresis loops to extract information for each magnetic phase. Finally, we investigate the magnetic interactions between the phases and verify the presence of exchange coupling between them. We observe the altering of the magnetic properties by the nanopores as a magnetic hardening of the magnetic material. The ZFC-FC (Zero-field cooled/field cooled) measurements indicate the presence of a disordered glass state below 50 K, which can be explained by the formation of iron oxide at the titanium oxide-iron interface with a short-range magnetic order.

## 1. Introduction

Nanoscale interface engineering of magnetic metals and transition metal oxide layers is of great interest due to their high spin polarization [1], voltage-controlled magnetization [2], magnetoelectric coupling [3], microwave absorption [4], and magnetoresistance [5], accompanied by a high Curie temperature of the material. Such a wide range of interesting physical properties makes these ferromagnetic heterostructures promising candidates for non-volatile memory applications, domain wall logic, and sensors [6]. However, the magnetic properties of the systems are significantly affected by the presence of different defects, such as grain boundaries, impurities, vacancies, non-magnetic inclusions, corrugations, or surface defects [7,8]. Typically, such defects are randomly created during thin film preparation, and therefore, they can alter the properties of the heterostructures in an uncontrolled manner. On the other hand, defects can also be introduced on purpose by ion irradiation [9,10], patterning, or template-assisted deposition [11,12] to enable the tuning of selected parameters in a controlled way and the creation of various magnetic nanostructures, such as nanotubes, nanowires, or antidots [13,14,15]. The defect interface engineering enabled by patterning makes it possible to obtain materials with novel properties which are not observed in epitaxial or highly crystallized thin films.

An efficient and straightforward way of introducing a network of artificial defects is anodization, which can provide an ordered array of nanopores or nanotubes with well-controlled size, distribution, and configuration. Magnetic films deposited on such arrays exhibit distinct properties from their flat counterparts [16,17,18]. Therefore, the introduction of the nanopores is an efficient method for the modification of ferromagnetic films and for the improvement of their magnetoresistance properties, magnetization reversal parameters, permeability, effective anisotropy [19,20,21,22], and coercivity due to the formation of pinning sites altering the domain wall movement [23,24,25].

One of the most common approaches to investigate the influence of artificial defects on magnetic properties of thin films is their deposition on anodized aluminum oxide (AAO) [26,27]. AAO is used in nanoscale interface engineering due to its ability to provide a large area matrix with a large range of accessible pore sizes [28]. High quality ordered arrays of nanostructures can also be fabricated on various transition metals, such as the versatile titanium. The anodized titanium oxide (ATiO) can grow in a form of nanotubes or nanopores, depending on the anodization parameters. The diameter of these nanostructures can be tuned in a range of 10 to 200 nm, while their length can vary from dozens of nanometers up to 10 μm [29]. Additionally, unlike AAO, ATiO is an n-type semiconductor with an energy gap of about 3 eV for rutile, which, in combination with a ferromagnetic layer, enables the creation of a magnetically controlled junction with spin-dependent transport properties [30]. The presence of spin-dependent characteristics in this type of system was shown by Sarkar et al. [31], while the possibility of a high polarization ratio of the injected current was demonstrated in patterned ferromagnetic/semiconductor layers by Roundy et al. [32]. The magnetic characteristics of such systems are significantly affected by the morphology of the magnetic layer interface. In particular, the magnetic properties, performance, and spin-polarization ratios of such systems are strongly influenced by surface imperfections and the intermixing of atoms since they induce changes in the distribution of local electric field and the demagnetization factor [32]. Understanding and investigating the impact of defects and interfaces on the magnetic properties of ferromagnetic/semiconductor layers is key for the future spintronic applications and devices based on these types of junctions. Such systematic study can be performed by introducing a well-controlled network of artificial defects into the model ferromagnetic/semiconductor system, e.g., Fe/ATiO. However, studies of such systems are rare and do not take into account the influence of thermal treatment, which is a common stage in the production of semiconducting junctions [33,34].

In this paper, we study the magnetic properties of iron thin films deposited on anodized titanium oxide with various characteristics of surface morphology and fabrication conditions. The ability to control the surface morphology of the titanium oxide by choice of anodization parameters allowed us to adjust the defect density, size, and roughness of the iron thin film layer deposited on the ATiO matrix. This gave us an opportunity to determine the correlation between the structure parameters and magnetic characteristics, and to identify what kind of alterations in magnetic properties originate from the interface. In particular, we explored the modifications of magnetic ordering and magnetic reversal caused by artificial defects and thermal treatment. Additionally, the inclusion of mesoscopic defects inducing intermixing of atoms at the ATiO/Fe interface led to the observation of distinct magnetic phases tuned by the size of nanopores in ATiO. We then used the modified Takács model [35] to extract information concerning saturation magnetization and coercivity for each magnetic phase, while the intrinsic magnetic properties of films were determined by zero-field cooled (ZFC) and field-cooled (FC) magnetization curves.

## 2. Materials and Methods

Multilayers were deposited on single crystal Si (100) substrates with sizes of 17 mm × 17 mm. Prior to the deposition process, the substrates were ultrasonically cleaned in acetone, isopropanol, and distilled water for 10 min for each step. Afterward, Ti (50 nm)/Au (100 nm)/Ti (300 nm) layers were deposited by e-beam evaporation in a vacuum chamber under the pressure of 10^−5^ mbar (Univex 300, Leybold, Koln, Germany). The bottom titanium layer served as an adhesive layer, the 100 nm of gold was the electric contact, and the top titanium layer was intended for nanopatterning and was deposited through a round mask with a diameter of 10 mm positioned at the center of the substrate.

The TiO_x_ nanostructures were fabricated by anodization using the two-electrode set-up, where deposited samples and platinum foil served as the working and counter electrodes, respectively. Both electrodes were connected to a direct current (DC) power supply (Delta Elektronika SM 300-10D, Delta Elektronika, Zierikzee, The Netherlands) and the working distance between them was 3 cm. The anodization chamber was filled with an electrolyte composed of glycerin (Glycerin anhydrous 99.5%, Chempur, Piekary Śląskie, Poland) and 0.5 wt% NH_4_F (Ammonium Fluoride 98%, Merck KGaA, Darmstadt, Germany). The anodization time was 45 min and the applied potentials were 5 V, 15 V, and 60 V. After anodization, the samples were rinsed with isopropanol and distilled water, and dried in a stream of hot air.

After the anodization, 50 nm of iron was deposited on the samples and further covered by 50 nm of a gold protective layer. The deposition was performed through a round mask with a diameter of 8 mm positioned at the center of anodized titanium oxide, while the remaining conditions were the same as previously described. Sample preparation procedure was completed by annealing at 450 °C for 15 min in a vacuum with a pressure of 10^−5^ mbar.

The surface morphology of the layers was studied by scanning electron microscopy (SEM, Tescan Vega 3, Tescan, Brno, Czech Republic) in secondary electron detection mode with the energy of the primary beam equal to 5 keV. The crystal structure was studied by X-ray diffraction (XRD, PANalytical X’Pert Pro, Malvern Panalytical, Malvern, UK) with Cu Kα radiation (λ = 1.5418 Å) and carried out in *θ*–2*θ* geometry. The measurements were conducted in the 2*θ* range from 20 to 90 degrees. Material composition was studied by the depth profiling carried out by secondary neutral mass spectroscopy measurement (SNMS, INA-X, SPECS, Berlin, Germany). The sputtered area was confined to a circle of 1 mm in diameter by a tantalum mask, and the sputtering was performed with an argon plasma after applying a negative voltage of 350 V. X-ray photoemission spectroscopy (XPS) was performed using an aluminum Kα source (10 kV accelerating voltage and 10 mA emission current) and a hemispherical energy analyzer (type Phoibos 100, SPECS, Berlin, Germany). The base vacuum in the instrument was 10^−10^ mbar. The magnetic measurements were conducted using a Magnetic Property Measurement System (MPMS XL SQUID, Quantum Design, San Diego, CA, USA) at room temperature and 10 K with the external magnetic field, up to 50 kOe, applied parallel (in-plane geometry, IP) or perpendicular (out-of-plane geometry, OOP) to the sample surface. Zero-field cooled/field cooled (ZFC-FC) measurements were performed in a field of 100 Oe in the temperature range from 5 K to 300 K. In the ZFC-FC measurement, the samples were first cooled from 300 K to 5 K without a magnetic field, and then a small magnetic field of 100 Oe was applied. After that, the ZFC magnetization (*M*_ZFC_) was recorded during heating from 5 K to 300 K. Subsequently, the sample was cooled to 5 K with the unchanged magnetic field, and the FC magnetization (*M*_FC_) was measured while cooling the system from 300 K to 5 K.

## 3. Results and Discussion

### 3.1. Structural Characterization

SEM images of titanium oxide surfaces obtained after anodization at different potentials are shown in Figure 1. The smallest anodization voltage of 5 V (Figure 1A) resulted in the fabrication of the titanium oxide in the form of granules, with an average size of approximately 100 nm. These granules are randomly distributed over the surface of the whole sample and create a mesoporous titanium oxide layer. Higher anodization voltages of 15 V and 60 V are required to create the matrix of nanoporous oxide, as shown in Figure 1B,C, respectively. The pores have either circular or elliptical shapes and are uniformly distributed on the surface. The analysis of inner pore sizes and surface porosity was performed with the ImageJ software (version 1.53f51) (see Figure 1D). The inner pore diameter *D*_p_ was calculated as a Feret diameter, while the porosity was determined as a ratio of the surface of the pores divided by the surface of the whole image.

Applying higher voltage results in the increase of the size of pores and porosity, which is a standard effect also observed in other anodization experiments [36]. Anodization at the chosen parameters resulted in the formation of nanoporous titanium oxide with mean pore diameters of 20 nm and 50 nm for the applied potentials of 15 V and 60 V, respectively. For the sample anodized at 5 V, the pores were too small to be observed, giving approximately zero porosity. The anodized samples were used as a substrate for the deposition of the 50 nm of iron subsequently covered with gold. The total thickness of the Fe and Au layers was high enough to fully cover and close the nanopores. For clarity of reading, the samples are named S*x*, where *x* is a value of the inner pore diameter.

To obtain structural information, the XRD measurements were carried out on both as-prepared and annealed samples. The diffraction measurements were performed with an offset equal to −5° to avoid the appearance of diffraction maxima from silicon single crystal substrate. Figure 2A presents representative diffraction patterns for the S20 sample. Other samples (S0 and S50) gave similar results. The bottom part of the graph shows the reference positions of gold and iron peaks based on ICDD datasheets no. 03-065-8601 and no. 01-085-1410.

The as-prepared sample S20 gave peaks corresponding to the *fcc* crystal structure of Au, with the most prominent peak at 38.4°, related to the (111) crystal plane. The reflections from iron (*bcc*) were also present, but they partially overlap with the gold peaks. The calculated lattice parameter for Au is *a* = 4.07 (2) Å and Fe is *a* = 2.86 (1) Å, which is in agreement with the bulk material values [37]. We also calculated the grain size of the iron. To calculate crystallite sizes *d*, the Scherrer equation, *d* = 0.9λ/βcos*θ*, was used, where *λ* is a wavelength, *β* is the line broadening at half the maximum intensity, and *θ* is the Bragg angle. The calculated grain size of the sample S20 was 27.6 (3.5) nm for the annealed sample and was almost two times larger than the deposition when grain size was 13.4 (2.8) nm. Grain sizes in other samples were similar to the S20 sample and also doubled after annealing.

The presence of titanium oxide in as-deposited samples was confirmed by two weak peaks from the rutile at 36.6° and 69.1°. The intensity of these peaks is low due to the weak crystallization of titanium oxide during the anodization process, which most often results in amorphous TiO_2_ [36].

The diffractogram of the annealed sample shows the change in the crystal structure of the oxide, confirmed by the appearance of diffraction peaks from various titanium oxides. The most intense diffraction maxima at 37.4° and 41.6° come from TiO and rutile [38], respectively. Both of them are related to the (111) crystallographic plane. The scarce amount of oxygen during the preparation process is responsible for the appearance of non-stoichiometric titanium oxide exhibiting diffraction peaks at 20.9°, 28.7°, 39.9°, and 55° [39,40]. The presence of titanium oxides with a low Ti oxidation state is a consequence of the lack of oxygen during annealing and the usage of electrolytes without water. Additionally, the oxygen atoms can diffuse through the ATiO/Fe interface and form iron oxides, the presence of which is suggested by the diffraction maximum found near 55°. However, the presence of iron oxide cannot be clearly identified since the diffraction peaks overlap with peaks coming from titanium oxide. Another effect caused by annealing is a shift of XRD peaks from gold to higher angles, caused by the shortening of the lattice parameter *a* from 4.07 (2) Å to 4.01 (2) Å. The compression is a consequence of the gold layer intermixing with iron atoms [41], noticeable on SNMS depth profiles (Figure 2B). The decrease of lattice parameters of gold results from the placing of the Fe atoms in the Au lattice. The iron atoms have a smaller atomic radius (124.1 pm) compared to gold atoms (144.2 pm), causing the decrease of the lattice plane distance of Au (111) [42]. While the as-prepared sample displays a sharp step between gold and iron layers, in the annealed sample, the iron atoms are mixed evenly within the gold layer. There is no evident acute step on the depth profile, but rather a smooth passage between Au and Fe layers with gold concentration decreasing inside the iron layer. On the other hand, the lack of a sharp step between Fe and ATiO in both the as-prepared and annealed samples is due to the porous structure of titanium oxide. The deposited iron partially penetrates the pores and shows a smooth transition in SNMS analysis for both Fe and Ti atoms.

### 3.2. Magnetic Properties

The magnetic hysteresis loops of the as-prepared and annealed S0 samples are presented in Figure 3A,B, respectively. Upper parts of the graphs show the first derivative of d*M*(*H*)/d*H*. In the case of the as-prepared sample, a single narrow peak with a maximum at 110 Oe is noticeable, while after annealing, an asymmetric peak is present, indicating a more complicated magnetic system. To deconvolute the magnetic signal, we adopted the modified Takács model T(*x*) [35,43]. The T(*x*) expression for the symmetric hysteresis loops with *n* ferromagnetic components is presented below:(1)$$T(x)=M(H)=\sum_{i=1}^{n}2M_{a}^{i}/\pi \;\left[atan\left(H\pm H_{c}^{i}/a^{i}\right)\;\right]\pm b^{i}+M_{b}^{i}\left[\;coth\;\left(\frac{H}{c^{i}}\right)-\frac{c^{i}}{H}\right]$$
where $b^{i}=\frac{M_{a}^{i}}{\pi }\left[\left(\frac{H_{m}^{i}-H_{c}^{i}}{a^{i}}\right)-\left(\frac{H_{m}^{i}+H_{c}^{i}}{a^{i}}\right)\right]$.

In this case, the saturation magnetization of the *i*-th component *M*_s_*^i^* is the sum of ferromagnetic *M*_a_*^i^* and paramagnetic *M*_b_*^i^*, which is part of magnetization. Both terms have different slopes with coefficients a and c, respectively. The horizontal shift of magnetization Mai is related to the coercive field *H*_c_*^i^*, and the vertical shift b is determined for the condition that the two branches of the loop coincide at the point corresponding to the field *H*_m_ for which they are saturated: *M*↑(*H*_m_) = *M*↓(*H*_m_) [44].

The as-prepared sample S0 shows typical behavior for the iron film with in-plane magnetic anisotropy, that can be modelled with *n* = 1, due to the presence of a single d*M*/d*H* peak with the narrow switching field distribution originating from the iron layer (see Figure 3A). After annealing, the d*M*/d*H* shows the presence of an additional magnetic component with the switching field at 250 Oe. The second magnetic phase (p2) appears due to oxidation of the iron at the ATiO/Fe interface (Figure 3B).

The introduction of nanopores results in the appearance of a second magnetic phase even before the thermal treatment (Figure 4). In this case, a phase with a higher *M*_s_ (p1) arises from the iron layer deposited on top of ATiO, while the second phase (with lower saturation magnetization (p2)) originates from the iron deposited inside the pores. Iron inside nanopores is partially oxidized because of the contact with titanium oxide and has limited space for growth, resulting in characteristics similar to those of the confined nanoparticles [45]. In the case of sample S20, the iron which gets inside the pores creates small and isolated magnetic grains, which result in the separated maxima on the d*M*/d*H* curve.

The increased pore size in sample S50 leads to a greater accumulation of the iron inside pores, seen as an increase in the contribution of a softer phase and an overlapping of maxima at the d*M*/d*H* curve (Appendix A). Annealing results in significant changes in the *M*(*H*) curves, such as a twofold increase of the coercivity in comparison with the as-prepared samples. The calculated amplification factor of coercivity caused by patterning and annealing is shown in Appendix A. The enhancement of *H*_c_ is accompanied by an increase in the contribution of the softer phase in a total magnetic signal, from 14% to 21% and from 39% to 55% for sample S20 and sample S50, respectively.

Figure 5 shows saturation magnetization *M*_s_ at 10 K for all samples, calculated as a percentage of saturation magnetization of bulk iron. The as-prepared sample S0 exhibits the largest saturation magnetization; however, the value is only 3/4 of the iron bulk magnetization. This difference may result from the microstructure of the thin film deposited on the porous substrate. As explained by Kim and Oliveria [46], the deposited Fe films are less dense than the bulk due to the mesoporous character of the layer, leading to the reduction of *M*_s_ by undesired air oxidation. Here, the samples were covered with 50 nm of a protective gold layer; therefore, oxidation through the upper interface of the iron layer should be excluded. However, the diffusion of oxygen atoms through anodized titanium oxide is still possible and leads to the formation of FeO at the ATiO/Fe interface, which was confirmed by XPS analysis (see Appendix A). The increase of working surface by patterning facilitates further oxidation of iron at the ATiO/Fe interface, resulting in the decrease of saturation magnetization for samples with wider nanopores.

After annealing, the saturation magnetization shows an almost linear decrease with increasing inner pore diameters. We can assume that the passivation of the iron layer leads to a similar thickness of iron oxide in each sample; however, the volume of FeO is larger for wider pores due to the larger contact area between ATiO and Fe. As a result, the wider the pores, the smaller the resulting saturation magnetization. We found similar linear behavior for the effective anisotropy constant, where a decrease of *K*_eff_ was present with an increasing size of the pores reflecting the influence of the non-planar morphology of the substrate (see Appendix A). The annealed nanoporous samples exhibit a rise of *M*_s_ when compared to the as-prepared systems, which is caused by the increase of crystallinity degree and the size of iron crystallites (in agreement with [47]) two times larger after heat treatment.

The behavior of saturation magnetization of sample S0 after annealing is different than for nanopatterned samples. As shown in Figure 5, *M*_s_ is larger for the as-deposited sample S0 and drops down after thermal treatment. The results indicate a lower degree of oxidation in the sample S0 compared to the nanoporous sample. The formation of iron oxide results from the diffusion of oxygen atoms at the ATiO/Fe interface; therefore, the amount of iron oxide depends on the contact area between the layers and is the lowest for the as-deposited S0 sample. During annealing, the diffusion leads to partial oxidation of iron at the ATiO surface, with the resultant thickness of iron oxide limited by the passivation thickness. Therefore, annealing results in an increased amount of iron oxide that is dependent on the size of the nanopores.

The evolution of coercive fields determined from the T(*x*) model, with respect to the inner pore diameter, is shown in Figure 6. The as-prepared sample S0 exhibits a value of *H*_c_ equal to 106(10) Oe, comparable to the *H*_c_ values for polycrystalline iron films with a similar thickness (about 50 nm) [48,49]. Annealing of the sample doubles the value of total coercivity and causes the appearance of the second magnetic phase. Structural defects, such as grain boundaries, together with surface roughness and misalignment of the magnetization axis between grains, have the largest impact on coercivity in polycrystalline films [50,51]. Heat treatment induces recrystallization, leading to the improvement of crystal structure and the decay of structural defects, hence, the coercivity should be smaller. However, the formation of iron oxide at the interface between ATiO and Fe leads to the appearance of a second, softer magnetic phase and an increase in coercivity. The rise of the coercive field is a result of inter-granular exchange hardening between hard and soft magnetic phases, as observed in the literature [52,53]. The heat treatment of the sample leads to the increase of iron oxide and, consequently, to an increase of contact area between the oxide and iron layer. This means more active sites for the intergranular exchange harden due to the increase of pinning sites and the roughness of the interface. In the case of patterned samples, the magnetic two-phase composition exists even before heat treatment. The values of the coercive field for S20 and S50 samples are higher than for the mesoporous specimen. This effect results from the pinning of magnetic domain walls at pores acting as artificial defects [54].

An exception is a soft phase of the as-prepared sample S20, where *H*_c_ is the lowest in the whole series. This is likely a consequence of the small amount of iron assembled inside the pores forming isolated, weakly ferromagnetic nanocrystallites of partially oxidized iron. Annealing of nanoporous samples fully oxidizes iron accumulated inside pores and, again, due to the inter-granular exchange hardening reversal of magnetization, is more energetically demanding, which leads to increased coercivity.

We performed recoil curve measurements to test the hypothesis about the specific behavior of the S20 sample and about its exchange interactions between different magnetic phases. Figure 7 shows the in-plane hysteresis curves measured for sample S20 at 10 K directly after deposition and after annealing. A comparison of the curves measured for both IP and OOP geometries is included in Appendix A, and shows that the easy axis of magnetization lies in the film plane. Similar behavior was found for samples S0 and S50. The in-plane anisotropy for the polycrystalline thin iron film is a typical property and has been observed for Fe films with similar thickness deposited on various substrates like glass, silicon, or Kapton [55,56]. Hysteresis loops for as-prepared and annealed samples display steps around *H* = 15 Oe and *H* = 200 Oe, respectively (see Figure 7). This indicates the existence of a soft magnetic phase that could be coupled to the harder magnetic phase by a weak exchange interaction [57,58]. To confirm the presence of the magnetic exchange coupling between soft and hard phases, we measured minor hysteresis curves, or recoil curves, during the reversal process shown in the insets of Figure 7. For a chosen reversal field *H*_R_, the magnetic field is reduced to 0 and increased again to the higher *H*_R_ value when the process is repeated. In the case of a strongly coupled system, the recoil curves remain closed since all magnetic moments show collective behavior, demonstrated, for example, in FeCo/FePt bilayer systems [59] or in PrFeB permanent magnets [60]. Open recoil curves are related to uncoupled soft and hard magnetic phases where the openness—the difference between the upper and lower branch of recoil curves at their widest point—depends on the strength of the magnetic interaction between soft and hard phases. Openness decreases with the increase of the strength of magnetic exchange interactions [58,61]. In the case of the as-prepared sample S20, a narrow openness of recoil curve for reversal field *H*_R_ of 350 Oe is found, indicating the existence of interphase magnetic exchange coupling [57]. The annealing causes the increase of the openness of recoil curves due to the increase in the soft phase contribution to a magnetic signal. This corresponds to the increased amount of material responsible for the soft phase, similar to the effect observed in iron-based nanocomposites [62].

To investigate the essential magnetic properties of nanoporous samples, we carried out the temperature dependent zero-field cooled and field cooled magnetization measurements. The ZFC and FC magnetization curves for sample S20 are shown in Figure 8. The ZFC-FC curves for the as-prepared sample demonstrate the dependence of ferromagnet below the Curie temperature of unsaturated magnetic moments. On the contrary, the curves for the annealed sample present well-defined maxima near 50 K, labelled as freezing temperature *T*_f_ on both *M*_ZFC_ and *M*_FC_ curves. The presence of the maxima is related to the appearance of iron oxides at the ATiO/Fe interface and the collective freezing of their magnetic moments [63], while at high temperatures, a ferromagnetic signal from the Fe layer dominates.

The presence of a low-temperature maximum *T*_f_ and a decrease of *M*_FC_, when lowering temperature below *T*_f_, suggests a presence of magnetic glass state, i.e., spin-glass or super spin-glass [64]. The possibility of the magnetic glass state can originate from the nanocrystallites of iron oxide formed inside nanopores. As we showed for recoil curve measurements of the sample S20, two weakly-coupled magnetic phases were observed. This means that part of iron oxide nanocrystallites can remain magnetically uncoupled and act as nanoparticles in the blocked state [65]. If, on the other hand, superparamagnetism is present in the system, it would lead to the increase of magnetization *M*_FC_ with a decreased temperature. The low-temperature behavior of the *M*_FC_ resembles the spin-glass-like state, which may arise from the iron oxide layer or disordered spins located at the grain boundary of ferromagnetic components [66,67,68].

## 4. Conclusions

We investigated the magnetic properties of iron thin films deposited on nanoporous titanium oxide as a function of inner pore diameter. The studies were performed directly after deposition and after thermal treatment. For annealed samples, the saturation magnetization and effective anisotropy constant were inversely proportional to the diameters of the pores, showing strong influence of the ATiO morphology. The magnetic hysteresis curves show the presence of two magnetic phases originating from the iron layer and the iron oxides formed at the ATiO/Fe interface. Based on the example of the as-deposited sample S20 and the magnetic recoil curves investigation, a weak exchange coupling interaction between iron and iron oxides was found and persisted even after annealing. The magnetic parameters of both phases were determined using the modified Takács model. The coercive field presented a significant enhancement up to four times, induced by nanoporous morphology and annealing of the samples. The evidence of the existence of a low-temperature glass-like magnetic state with a freezing temperature *T*_f_ ≈ 50 K was shown; it originates from the chemical disorder and distribution of oxidized iron nanocrystallite sizes formed at the ATiO/Fe interface.

## Figures and Tables

**Figure 1 materials-16-00289-f001:**
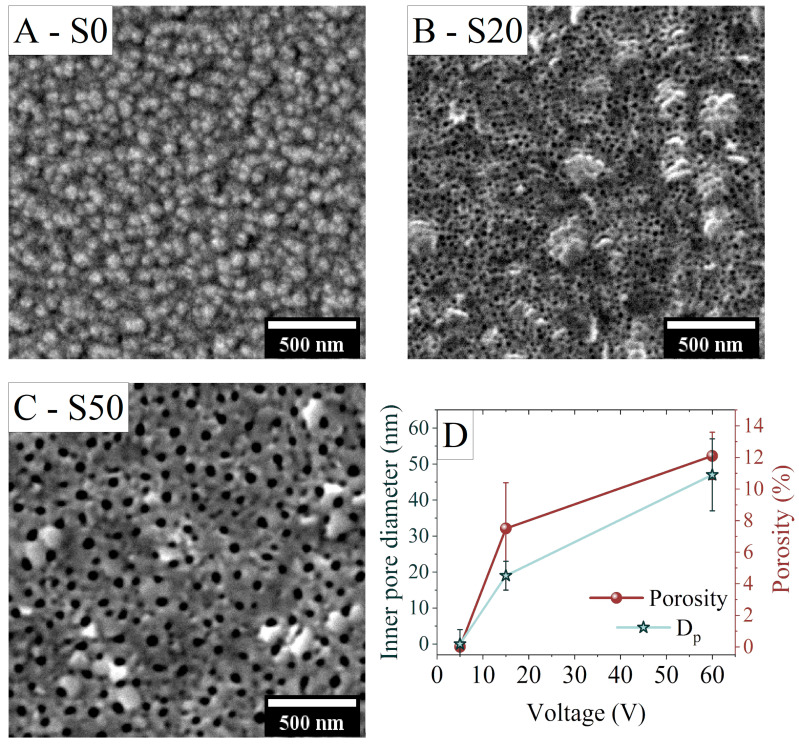
The morphology of anodized titanium oxide obtained at (**A**) 5 V, (**B**) 15 V, and (**C**) 60 V. The voltage impact on the inner pore diameter and the porosity is shown in (**D**).

**Figure 2 materials-16-00289-f002:**
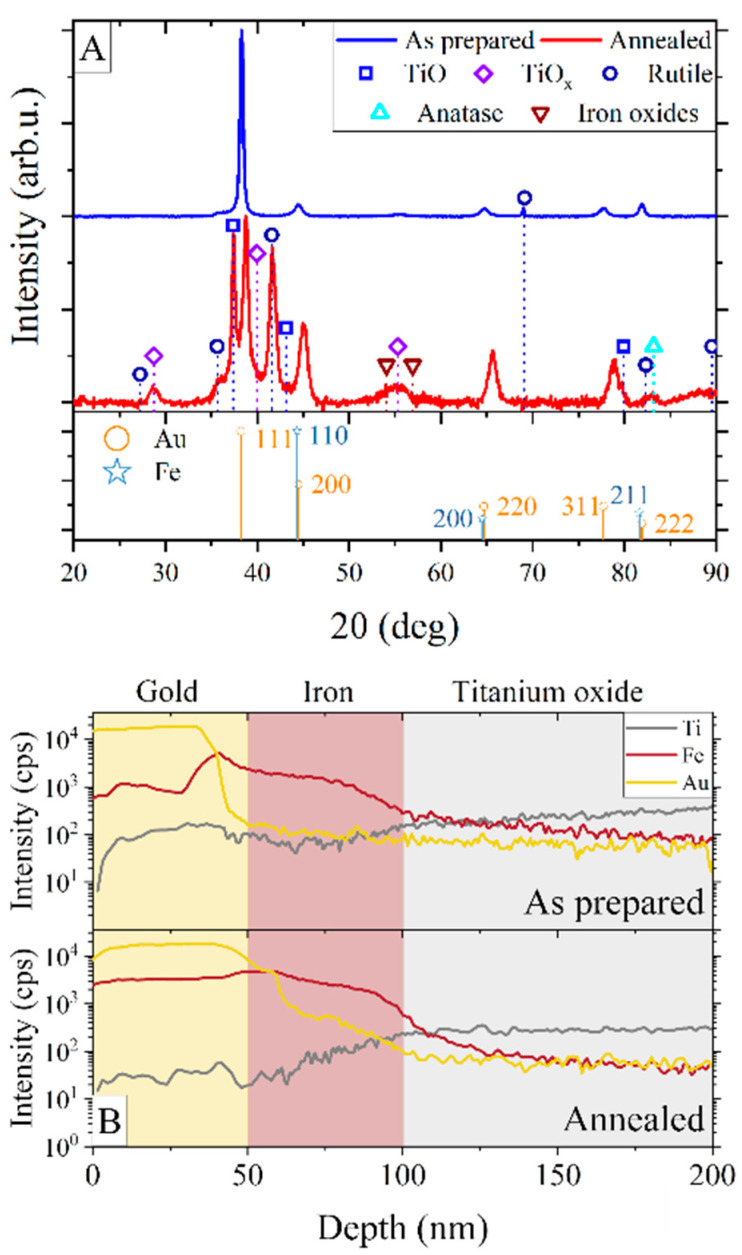
XRD patterns (**A**) and SNMS depth profiles (**B**) for as-prepared and annealed S20 samples.

**Figure 3 materials-16-00289-f003:**
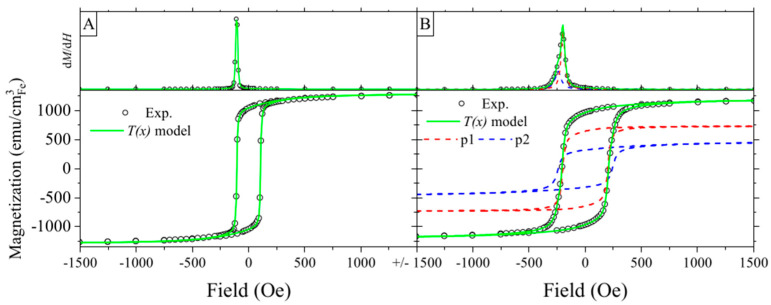
Magnetization curves (bottom graphs) for the S0 sample as-prepared (**A**) and annealed (**B**), and derivatives of the upper branch of hysteresis loops (top graphs). Calculated hystereses of different magnetic phases are shown with dashed lines. Loops were measured in the in-plane geometry.

**Figure 4 materials-16-00289-f004:**
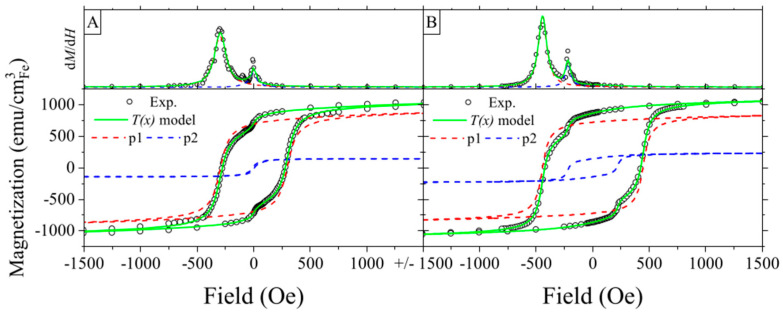
Magnetization curves (bottom graphs) for the S20 sample as-prepared (**A**) and annealed (**B**), and derivatives of the upper branch of hysteresis loops (top graphs). Calculated hystereses of different magnetic phases are shown with dashed lines. Loops were measured in the in-plane geometry.

**Figure 5 materials-16-00289-f005:**
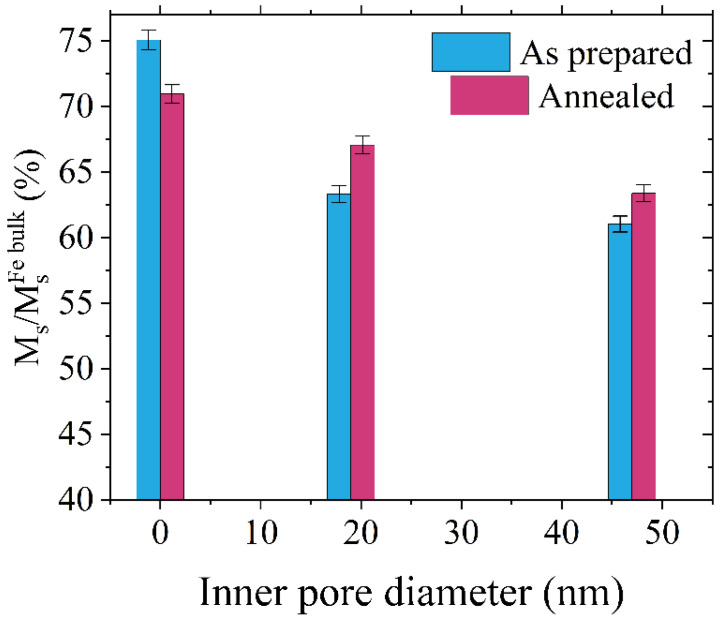
The saturation magnetization dependence on pore size for as-prepared and annealed samples. Bars show the comparison of the saturation magnetization of Fe/ATiO system to the saturation magnetization of bulk Fe.

**Figure 6 materials-16-00289-f006:**
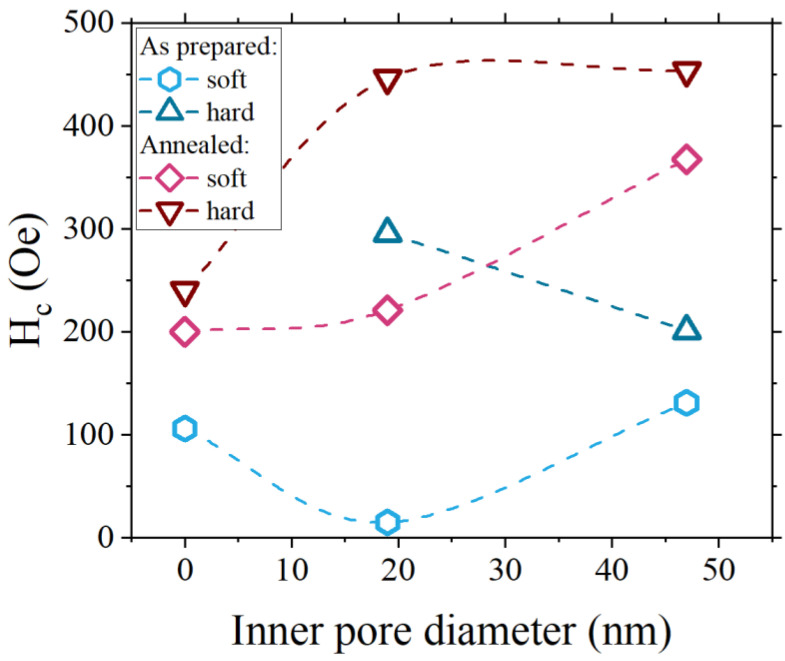
The 10 K coercive field of both magnetic phases with respect to the inner pore diameter.

**Figure 7 materials-16-00289-f007:**
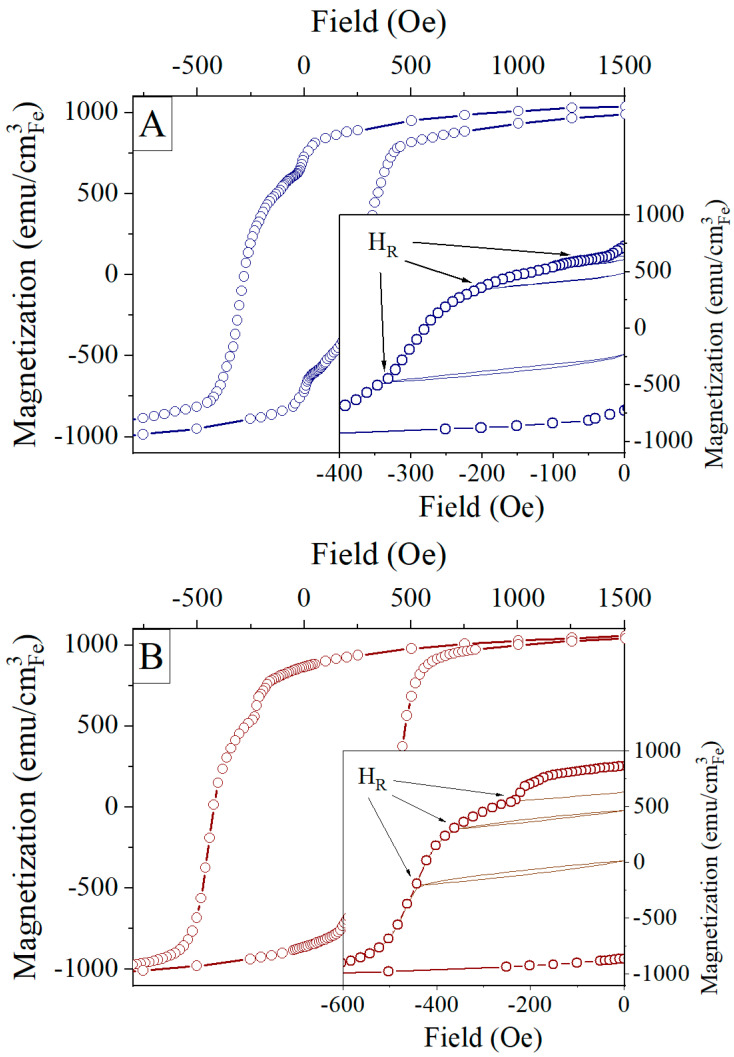
The hysteresis curves for sample S20 as-prepared (**A**) and annealed (**B**). The recoil curves (continuous lines) are shown in the insets. Magnetization and recoil curves were measured in the in-plane geometry.

**Figure 8 materials-16-00289-f008:**
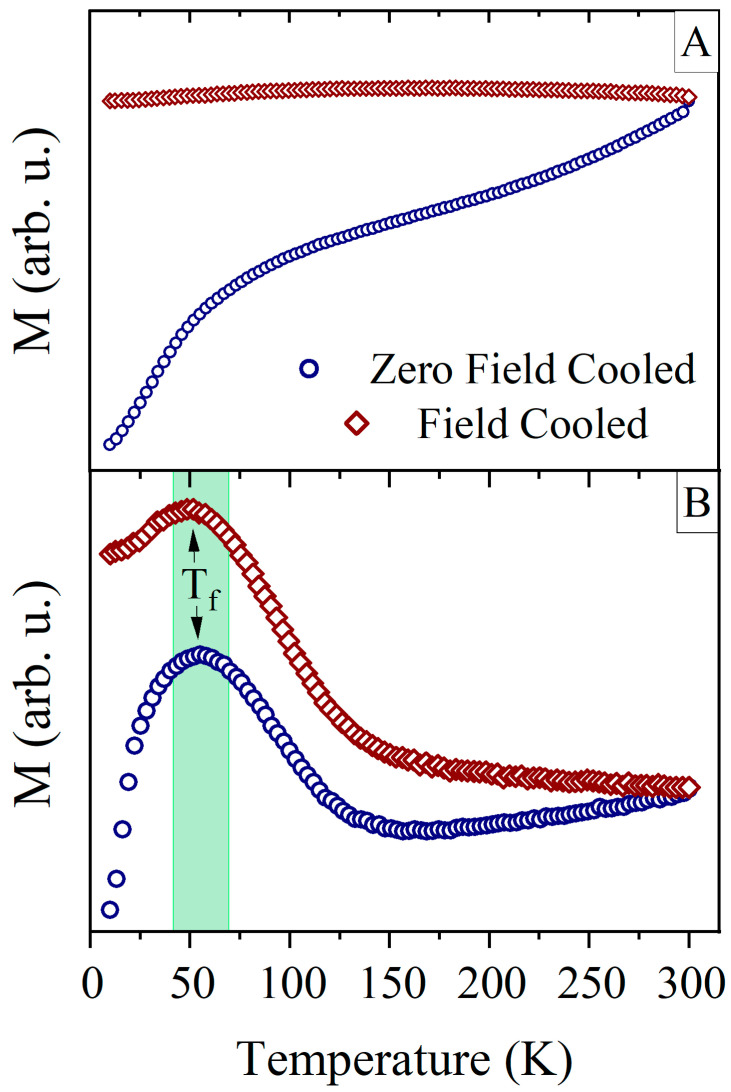
ZFC-FC magnetization curves for as-prepared (**A**) and annealed (**B**) sample S20.

## Data Availability

The data presented in this study are available on request from the corresponding author.

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
