# Peer review of "Tuning of the Titanium Oxide Surface to Control Magnetic Properties of Thin Iron Films"

_materials, 2022, doi:10.3390/ma16010289_

Round 1
Reviewer 1 Report
In their paper Chojenka et al. showed elegantly that modification of titanium oxide surface morphology via changing the anodization conditions could bring possibility to control magnetic properties of Fe-based thin films. The topic is relevant due to various potential applications of magnetically coupled multiphase structures. The conclusions are supported by experimental results received in systematical way by different structural, chemical and magnetic measurements. The manuscript is well written and presentation is clear. Thus, I recommend this paper for publishing in the present form.
Author Response
We thank the reviewer for his positive opinion of our work.
Reviewer 2 Report
Referee's report on the paper “materials-2074630: ”Tuning of the titanium oxide surface......, by J. Chojenka, et al..
The authors have investigated the magnetic properties of thin iron film deposited on anodized titanium oxide substrates, whose surface morphology was modified with suitable processing, causing saturation magnetization and magnetic hysteresis loops to depend strongly, in particular, on the porosity the substrate. Both structural and magnetic properties are investigated in detail by the authors. Regarding the magnetic properties, it is realized that porosity of the substrate leads to a decrease of the saturation magnetization and the emergence of a two-phase magnetic behavior of the film. The two phases are identified with the iron layer and the iron oxide layer formed upon annealing at the interface of iron with TiO, respectively. The hypothesis of weak exchange coupling between iron and iron oxide is put forward. An attempt to quantitatively identify the respective contributions of these two phases to the overall magnetic hysteresis behavior is made.
This work is focused on quite specific properties descending on interface morphology of thin layers, but it may eventually provide information having more general character. It is therefore of interest for scientists working in the area of thin magnetic films and multilayers. A few points, however, would call, in my opinion, for clarification.
1) The annealing process appears to lead, judging from the SNMS profiles shown in Fig. 2b, to a certain penetration of Fe into Au and viceversa. One might wonder about the volume you have taken in determining the magnetization of the sample in Figs. 3 and 4. Let me recommend here to use the SI units for the magnetic quantities (T and A/m).
2) You identify in Fig. 3b (annealed sample) the hysteresis loops of two different magnetic phases. Since no discontinuity apparently emerges in the loop profile, you invoke asymmetry of dM/dH as a revealing clue. However, this fact hardly indicates, in my opinion, the presence of two magnetic phases. In fact, asymmetry of dM/dH across the loop branch is a universal feature of all ordinary single-phase magnetic materials.
3) You stress an apparent incongruence regarding the increase of the coercive field of the sample upon annealing (page 9). It is not clear to me how the appearance and growth of a second soft magnetic phase might eventually induce an increase of coercivity. It is true that, given the involved layer thickness, exchange hardening can occur across the interface. What is not clear (that is, unexplained by the authors) is why the iron phase should suffer hardening, in spite of the structural improvement due to annealing.
4) How could you find the effective anisotropy constant? Its trend versus nanopore size appears to be opposite to the trend shown by the coercive field, that is, at odds with the conventional expectation of coercivity and anisotropy going hand-in-hand.
Author Response
This work is focused on quite specific properties descending on the interface morphology of thin layers, but it may eventually provide information having a more general character. It is therefore of interest for scientists working in the area of thin magnetic films and multilayers. A few points, however, would call, in my opinion, for clarification.
1) The annealing process appears to lead, judging from the SNMS profiles shown in Fig. 2b, to a certain penetration of Fe into Au and viceversa. One might wonder about the volume you have taken in determining the magnetization of the sample in Figs. 3 and 4. Let me recommend here to use the SI units for the magnetic quantities (T and A/m).
We would like to thank the reviewer for the comment. In all cases, the magnetization values were normalized to the total volume of the iron which do not change within the samples after annealing. Considering the conversion to SI units we would prefer not to do it. In magnetism both unit systems, SI and cgs are equally used due to historical and practical reasons, most of the instruments are calibrated and operate in cgs units. This fact results in the use by the community of both systems of units.
2) You identify in Fig. 3b (annealed sample) the hysteresis loops of two different magnetic phases. Since no discontinuity apparently emerges in the loop profile, you invoke asymmetry of dM/dH as a revealing clue. However, this fact hardly indicates, in my opinion, the presence of two magnetic phases. In fact, asymmetry of dM/dH across the loop branch is a universal feature of all ordinary single-phase magnetic materials.
We agree with the reviewer that asymmetry of the dM/dH could originate from the imperfection of the material and the irreversibility part of magnetization in single-phase magnetic material. But in our case, the asymmetry is an effect of the formation of iron oxide at the interface. According to the XRD diffractogram and XPS spectra measured at the interface, we observed the formation of iron oxide after the annealing, including the S0 sample. Knowing the phase composition of the annealed sample, we have modeled its magnetization curve with the T(x) model for two magnetic phases to calculate the contribution of iron and iron oxide in the total magnetization signal.
3) You stress an apparent incongruence regarding the increase of the coercive field of the sample upon annealing (page 9). It is not clear to me how the appearance and growth of a second soft magnetic phase might eventually induce an increase of coercivity. It is true that, given the involved layer thickness, exchange hardening can occur across the interface. What is not clear (that is, unexplained by the authors) is why the iron phase should suffer hardening, in spite of the structural improvement due to annealing.
The annealing causes two effects: one is the structural improvement of the iron, which should decrease the number of intrinsic defects, and the second is the oxidation of the Fe layer resulting in the formation of the iron oxide at the ATiO/Fe interface. The heat treatment leads to the increase of iron oxide and in consequence, an increase of contact area between the oxide and iron layer. It induces more active sites for intergranular exchange hardening due to the increase of pinning sites and roughness of the interface. We thank the reviewer for pointing out this problem, and we have included this explanation in the manuscript (line 316)
4) How could you find the effective anisotropy constant? Its trend versus nanopore size appears to be opposite to the trend shown by the coercive field, that is, at odds with the conventional expectation of coercivity and anisotropy going hand-in-hand.
The effective anisotropy constants were determined from the difference between the area under the demagnetization branch of M(H) taken for the in-plane and out-of-plane geometry (easy and hard magnetization direction) according to the formula given in the supplementary materials.
The increase of the coercive field with the nanopore size is caused by the pinning of the magnetic moments on the defects, their number increases with the increase of nanopore diameter, while the effective anisotropy constant originates mainly from the shape anisotropy of the iron and iron oxide layers. The shape anisotropy strongly depends on the sample topography as well as on the roughness of the layer. We expect a decrease in effective anisotropy constant due to the increase of the contribution of the iron oxide in the samples for samples with larger pores. The increased amount of iron oxide with the size of pores is observed as a decrease of the magnetic saturation in Fig. 5. Taking into account these two aspects a decrease in the Keff is expected.
Reviewer 3 Report
The authors present an exhaustive study on magnetic exchange between Fe/ATiO thin films.
My observations:
1. What is the anodized voltage limit which decides the nanopores. To put it other way, how is the optimization on nanopore size decided for this study?
2. The authors mention "Iron inside nanopores is partially oxidized because of contact with titanium oxide and has limited space for growth resulting in characteristics similar to those of confined nanoparticles." Request the authors to present concrete evidence (EDX or others) to justify the claim.
3. Can the authors throw light on any domain wall interactions between Ti and Fe at the interface.
Author Response
The authors present an exhaustive study on magnetic exchange between Fe/ATiO thin films.
My observations:
- What is the anodized voltage limit which decides the nanopores. To put it other way, how is the optimization on nanopore size decided for this study?
The limit of the voltage for the size of nanopores strongly depends on the composition of the electrolyte and chosen substrate. According to the literature, anodization of the aluminum or titanium foil can be performed at voltages up to 180 V and 100 V, respectively. However, the anodization of the thin films is a complex process that requires more caution to prevent the detachment of the layer from the sample surface during the anodization process.
For layer stacking of Si/Ti(50 nm)/Au(100 nm)/Ti (300 nm), the optimal anodization parameters and electrolyte composition were experimentally obtained based on the previous experience of the research team. First, we oxidized the titanium layer using an electrolyte based on ethylene glycerol, however, the quality of obtained samples was not satisfactory due to their high fragility. To improve the quality of the titanium oxide layer, we decided to slow down the speed of oxidation in the electrolyte based on glycerin. It resulted in the nanoporous titanium oxide being of good quality as shown in the manuscript. The maximum voltage used for the anodization of samples was 60 V. Above this value the process became too rapid to be controlled leading to the delamination of the formed titanium oxide layer. The optimization of the nanopore manufacturing was decided through an iterative process of the creation of numerous samples and validation of their quality.
- The authors mention "Iron inside nanopores is partially oxidized because of contact with titanium oxide and has limited space for growth resulting in characteristics similar to those of confined nanoparticles." Request the authors to present concrete evidence (EDX or others) to justify the claim.
The oxidation of the iron layer at the surface of the anodized titanium oxide was confirmed by the XPS measurements after argon sputtering of the Fe/Au in the top layers during SNMS measurements. The XPS spectrum of the Fe/ATiO interface is shown in the Supplementary Materials (Figure S3). Analysis of this spectrum revealed that the majority of the iron at the Fe/ATiO interface was oxidized.
- Can the authors throw light on any domain wall interactions between Ti and Fe at the interface.
The nanopatterned titanium oxide, in normal conditions, is a paramagnetic material. The interactions at the Fe/ATiO interface can occur between the domain walls of iron and the nanopores. In this case, the nanopores act as pinning sites for the domain walls, an effect commonly seen in ferromagnetic antidots. In order to show the domain's wall interactions with the nanopatterned substrates, it is necessary to perform micromagnetic simulations, which were not in the scope of our interest.
Reviewer 4 Report
The authors have studied the magnetic properties of thin iron films deposited on the nanoporous titanium oxide templates and analyzed the properties depending on nanopore radius. The authors also compared the magnetic properties of annealed and as-prepared samples using M(H) loop. However, to consider this article for publication, the authors should address the following issues. So, I am recommending the manuscript for minor corrections.
Comments to the author
1. Please carefully proof-read and spell check manuscript to eliminate grammatical errors.
2. The author have mentioned “The ZFC-FC curves for the as-prepared sample demonstrate the dependence as for typical ferromagnetic material with the unsaturated magnetic moment.” How does the unsaturated magnetic moment account for the ferromagnetic properties?
3. The author has mentioned the possibility of spin glass state in the sample S20. The evidence of spin glass behavior is missing, and the author could have added the ZFC-FC curves for various magnetic fields and check whether the behavior is similar in those fields. AC susceptibility measurements are recommended for the confirmation of the spin glass behavior. (Mention the field in which the ZFC-FC measurement taken in the Figure 8)
Author Response
Comments to the author
- Please carefully proof-read and spell check manuscript to eliminate grammatical errors.
We carefully check our manuscript to make the necessary language editing with the help of a native speaker.
- The author have mentioned “The ZFC-FC curves for the as-prepared sample demonstrate the dependence as for typical ferromagnetic material with the unsaturated magnetic moment.” How does the unsaturated magnetic moment account for the ferromagnetic properties?
We thank the reviewer for bringing up this issue. We expressed in an unclear way that the temperature range of the ZFC-FC curve for the as-prepared sample was below the Curie temperature of the iron and we observed the behavior of ferromagnet in an unsaturated state since the magnetic field used during these measurements was 100 Oe, significantly lower than the saturation field of 10 kOe. We have changed the text accordingly.
The ZFC-FC curves for the as-prepared sample demonstrate the dependence of ferromagnet below Curie temperature of unsaturated magnetic moments.
- The author has mentioned the possibility of spin glass state in the sample S20. The evidence of spin glass behavior is missing, and the author could have added the ZFC-FC curves for various magnetic fields and check whether the behavior is similar in those fields. AC susceptibility measurements are recommended for the confirmation of the spin glass behavior. (Mention the field in which the ZFC-FC measurement taken in the Figure 8)
We agree with the reviewer that the statement given in the text indicates a potential hypothesis. The best way to verify would be to make AC magnetic measurements, but our samples have a considerable amount of paramagnetic conductors (gold and titanium layers) leading to the appearance of eddy currents, which perturbs the measured signal. The measurements of ZFC-FC DC magnetization in various fields are influenced by the ferromagnetic Fe layer which interferes with the signal from iron oxide. The presence of the signal from the FM layer makes it difficult to prove the existence of a glassy phase. We followed the explanation of Bedanta et al. (ref [66]) who unambiguously assigned the decrease of the field-cooled magnetization below freezing temperature to the spin-glass state. To resolve the problem we decided to modify this part of the text.
Round 2
Reviewer 3 Report
The revised manuscript can be accepted in its present form.